# Emergence of Segmentation with Minimalistic White-Box Transformers

Yaodong Yu[1,*] Tianzhe Chu[1,2,*] Shengbang Tong[1,3] Ziyang Wu[1] Druv Pai[1]

Sam Buchanan[4] Yi Ma[1,5]

[1]UC Berkeley  [2]ShanghaiTech  [3]NYU  [4]TTIC  [5]HKU

(*Equal contribution)

https://ma-lab-berkeley.github.io/CRATE

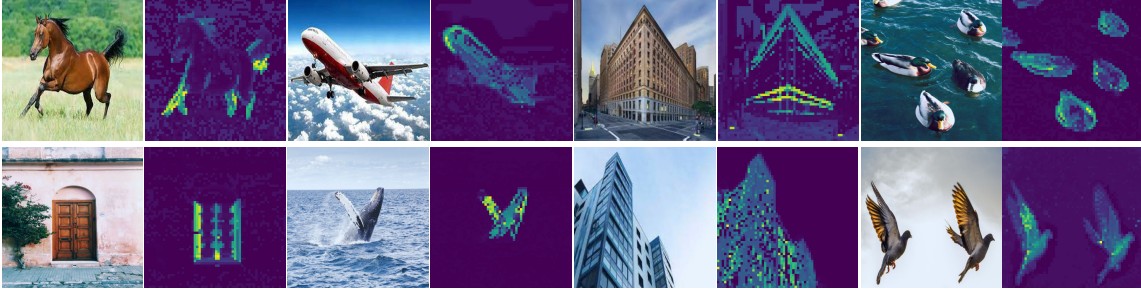

Figure 1: **Self-attention maps from a supervised CRATE with** $8 \times 8$ **patches** trained using classification. The **CRATE** architecture automatically learns to perform object segmentation without a complex self-supervised training recipe or any fine-tuning with segmentation-related annotations.

Transformer-like models for vision tasks have recently proven effective for a wide range of downstream applications such as segmentation and detection. Previous works have shown that segmentation properties emerge in vision transformers (ViTs) trained using self-supervised methods such as DINO, but not in those trained on supervised classification tasks. In this study, we probe whether segmentation emerges in transformer-based models *solely* as a result of intricate self-supervised learning mechanisms, or if the same emergence can be achieved under much broader conditions through proper design of the model architecture. Through extensive experimental results, we demonstrate that when employing a white-box transformer-like architecture known as CRATE, whose design explicitly models and pursues low-dimensional structures in the data distribution, segmentation properties, at both the whole and parts levels, already emerge with a minimalistic supervised training recipe. Layer-wise finer-grained analysis reveals that the emergent properties strongly corroborate the designed mathematical functions of the white-box network. Our results suggest a path to design white-box foundation models that are simultaneously highly performant and mathematically fully interpretable. Code is at https://github.com/Ma-Lab-Berkeley/CRATE.

## 1. Introduction

*Representation learning* in an intelligent system aims to transform high-dimensional, multi-modal sensory data of the world—images, language, speech—into a compact form that preserves its essential low-dimensional structure, enabling efficient recognition (say, classification), grouping (say, segmentation), and tracking [26, 31]. Classical representation learning frameworks, hand-designed for distinct data modalities and tasks using mathematical models for data [12, 38, 39, 48, 49], have largely been replaced by deep learning-based approaches, which train black-box deep networks with massive amounts of heterogeneous data on simple tasks, then adapt the learned representations on downstream tasks [3, 4, 35]. This data-driven approach has led to tremendous empirical successes—in particular, *foundation models* [3] have demonstrated state-of-the-art results in fundamental vision tasks such as segmentation [22] and tracking [45]. Among vision foundation models, DINO [6,

First Conference on Parsimony and Learning (CPAL 2024).

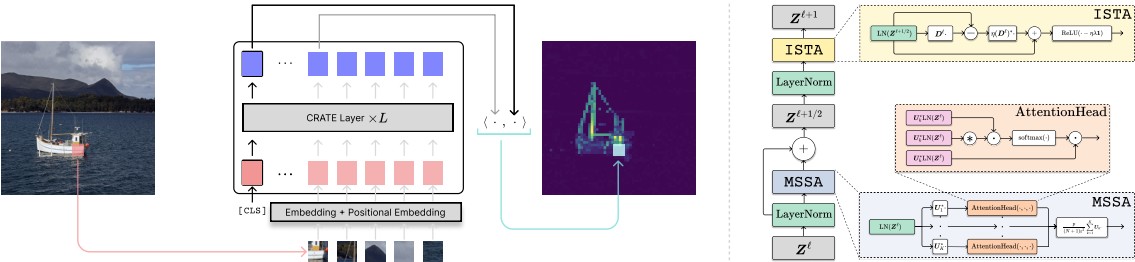

**Figure 2:** (*Left*) **Visualizing the self-attention map for an input image using the CRATE model.** The input tokens for **CRATE** consist of $N$ non-overlapping image patches and a `[CLS]` token. We use the **CRATE** model to transform these tokens to their representations, and de-rasterize the self-attention map associated to the `[CLS]` token and the image patch tokens at the penultimate layer to generate a heatmap visualization. Details are provided in Section 3.1. (*Right*) **Overview of one layer of CRATE architecture.** The **CRATE** model is a white-box transformer-like architecture derived via unrolled optimization on the sparse rate reduction objective (Section 2). Each layer compresses the distribution of the input tokens against a local signal model, and sparsifies it in a learnable dictionary. This makes the model mathematically interpretable and highly performant [51].

35] showcases a surprising *emergent properties* phenomenon in self-supervised vision transformers (ViTs [11])—ViTs contain explicit semantic segmentation information even without trained with segmentation supervision. Follow-up works have investigated how to leverage such segmentation information in DINO models and achieved state-of-the-art performance on downstream tasks, including segmentation, co-segmentation, and detection [2, 46].

As demonstrated in Caron et al. [6], the penultimate-layer features in ViTs trained with DINO correlate strongly with saliency information in the visual input—for example, foreground-background distinctions and object boundaries (similar to the visualizations shown in Figure 1)—which allows these features to be used for image segmentation and other tasks. However, to bring about the emergence of these segmentation properties, DINO requires a delicate blend of self-supervised learning, knowledge distillation, and weight averaging during training. It remains unclear if every component introduced in DINO is essential for the emergence of segmentation masks. In particular, there is no such segmentation behavior observed in the vanilla supervised ViT models that are trained on classification tasks [6], although DINO employs the same ViT architecture as its backbone.

In this paper, we question the prevailing wisdom, stemming from the successes of DINO, that a complex self-supervised learning pipeline is necessary to obtain emergent properties in transformer-like vision models. We contend that an equally-promising approach to promote segmentation properties in transformer is to *design the transformer architecture with the structure of the input data in mind*, representing a marrying of the classical approach to representation learning with the modern, data-driven deep learning framework. We call such an approach to transformer architecture design *white-box transformer*, in contrast to the black-box transformer architectures (e.g., ViTs [11]) that currently prevail as the backbones of vision foundation models. We experiment with the white-box transformer architecture **CRATE** proposed by Yu et al. [51], an alternative to ViTs in which each layer is mathematically interpretable, and demonstrate through extensive experiments that:

> The *white-box design of CRATE* leads to the emergence of segmentation properties in the network's self-attention maps, solely through a *minimalistic supervised training recipe*—the supervised classification training used in vanilla supervised ViTs [11].

We visualize the self-attention maps of **CRATE** trained with this recipe in Figure 1, which share similar qualitative (object segmentation) behaviors to the ones shown in DINO [6]. Furthermore, as to be shown in Figure 7, each attention head in the learned white-box **CRATE** seems to capture a different semantic part of the objects of interest. This represents the *first supervised vision model with emergent segmentation properties*, and establishes white-box transformers as a promising direction for interpretable data-driven representation learning in foundation models.

**Outline.** The remainder of the paper is organized as follows. In Section 2, we review the design of **CRATE**, the white-box transformer model we study in our experiments. In Section 3, we outline our experimental methodologies to study segmentation in transformer-like architectures, and provide a basic analysis which compares the segmentation in supervised **CRATE** to the vanilla supervised ViT and DINO. In Section 4, we present extensive ablations and more detailed analysis of the segmentation

property which utilizes the white-box structure of CRATE, and we obtain strong evidence that the white-box design of CRATE is the key to the emergent properties we observe.

**Notation.** We denote the (patchified) input image by $\boldsymbol{X} = [\boldsymbol{x}_1, \ldots, \boldsymbol{x}_N] \in \mathbb{R}^{D \times N}$, where $\boldsymbol{x}_i \in \mathbb{R}^{D \times 1}$ represents the $i$-th image patch and $N$ represents the total number of image patches. $\boldsymbol{x}_i$ is referred to as a *token*, and this term can be used interchangeably with image patch. We use $f \in \mathcal{F} : \mathbb{R}^{D \times N} \to \mathbb{R}^{d \times (N+1)}$ to denote the mapping induced by the model; it is a composition of $L + 1$ layers, such that $f = f^L \circ \cdots \circ f^\ell \circ \cdots \circ f^1 \circ f^0$, where $f^\ell : \mathbb{R}^{d \times (N+1)} \to \mathbb{R}^{d \times (N+1)}, 1 \leq \ell \leq L$ represents the mapping of the $\ell$-th layer, and $f^0 : \boldsymbol{X} \in \mathbb{R}^{D \times N} \to \boldsymbol{Z}^1 \in \mathbb{R}^{d \times (N+1)}$ is the pre-processing layer that transforms image patches $\boldsymbol{X} = [\boldsymbol{x}_1, \ldots, \boldsymbol{x}_N]$ to tokens $\boldsymbol{Z}^1 = [\boldsymbol{z}^1_{\texttt{[CLS]}}, \boldsymbol{z}^1_1, \ldots, \boldsymbol{z}^1_N]$, where $\boldsymbol{z}^1_{\texttt{[CLS]}}$ denotes the "class token", a model parameter eventually used for supervised classification in our training setup. We let

$$\boldsymbol{Z}^\ell = [\boldsymbol{z}^\ell_{\texttt{[CLS]}}, \boldsymbol{z}^\ell_1, \ldots, \boldsymbol{z}^\ell_N] \in \mathbb{R}^{d \times (N+1)} \tag{1}$$

denote the input tokens of the $\ell^{\text{th}}$ layer $f^\ell$ for $1 \leq \ell \leq L$, so that $\boldsymbol{z}^\ell_i \in \mathbb{R}^d$ gives the representation of the $i^{\text{th}}$ image patch $\boldsymbol{x}_i$ before the $\ell^{\text{th}}$ layer, and $\boldsymbol{z}^\ell_{\texttt{[CLS]}} \in \mathbb{R}^d$ gives the representation of the class token before the $\ell^{\text{th}}$ layer. We use $\boldsymbol{Z} = \boldsymbol{Z}^{L+1}$ to denote the output tokens of the last ($L^{\text{th}}$) layer.

# 2. Preliminaries: White-Box Vision Transformers

In this section, we revisit the CRATE architecture (Coding RAte reduction TransformEr)—a white-box vision transformer proposed in Yu et al. [51]. CRATE has several distinguishing features relative to the vision transformer (ViT) architecture [11] that underlie the emergent visual representations we observe in our experiments. We first introduce the network architecture of CRATE in Section 2.1, and then present how to learn the parameters of CRATE via supervised learning in Section 2.2.

## 2.1. Design of CRATE—A White-Box Transformer Model

**Representation learning via unrolling optimization.** As described in Yu et al. [51], the white-box transformer CRATE $f : \mathbb{R}^{D \times N} \to \mathbb{R}^{d \times (N+1)}$ is designed to transform input data $\boldsymbol{X} \in \mathbb{R}^{D \times N}$ drawn from a potentially nonlinear and multi-modal distribution to *piecewise linearized and compact* feature representations $\boldsymbol{Z} \in \mathbb{R}^{d \times (N+1)}$. It does this by posing a *local signal model* for the marginal distribution of the tokens $\boldsymbol{z}_i$. Namely, it asserts that the tokens are approximately supported on a union of several, say $K$, low-dimensional subspaces, say of dimension $p \ll d$, whose orthonormal bases are given by $\boldsymbol{U}_{[K]} = (\boldsymbol{U}_k)_{k=1}^K$ where each $\boldsymbol{U}_k \in \mathbb{R}^{d \times p}$. With respect to this local signal model, the CRATE model is designed to optimize the *sparse rate reduction* objective [51]:

$$\max_{f \in \mathcal{F}} \mathbb{E}_{\boldsymbol{Z}}\big[\Delta R(\boldsymbol{Z} \mid \boldsymbol{U}_{[K]}) - \lambda \|\boldsymbol{Z}\|_0\big] = \max_{f \in \mathcal{F}} \mathbb{E}_{\boldsymbol{Z}}\big[R(\boldsymbol{Z}) - \lambda \|\boldsymbol{Z}\|_0 - R^c(\boldsymbol{Z}; \boldsymbol{U}_{[K]})\big], \tag{2}$$

where $\boldsymbol{Z} = f(\boldsymbol{X})$, the coding rate $R(\boldsymbol{Z})$ is (a tight approximation for [30]) the average number of bits required to encode the tokens $\boldsymbol{z}_i$ up to precision $\varepsilon$ using a Gaussian codebook, and $R^c(\boldsymbol{Z} \mid \boldsymbol{U}_{[K]})$ is an upper bound on the average number of bits required to encode the tokens' projections onto each subspace in the local signal model, i.e., $\boldsymbol{U}_k^* \boldsymbol{z}_i$, up to precision $\varepsilon$ using a Gaussian codebook [51]. When these subspaces are sufficiently incoherent, the minimizers of the objective (2) as a function of $\boldsymbol{Z}$ correspond to axis-aligned and incoherent subspace arrangements [52].

Hence, a network designed to optimize (2) by unrolled optimization [7, 16, 32] incrementally transforms the distribution of $\boldsymbol{X}$ towards the desired canonical forms: each iteration of unrolled optimization becomes a layer of the representation $f$, to wit

$$\boldsymbol{Z}^{\ell+1} = f^\ell(\boldsymbol{Z}^\ell), \tag{3}$$

with the overall representation $f$ thus constructed as

$$f : \boldsymbol{X} \xrightarrow{f^0} \boldsymbol{Z}^1 \to \cdots \to \boldsymbol{Z}^\ell \xrightarrow{f^\ell} \boldsymbol{Z}^{\ell+1} \to \cdots \to \boldsymbol{Z}^{L+1} = \boldsymbol{Z}. \tag{4}$$

Importantly, in the unrolled optimization paradigm, *each layer $f^\ell$ has its own, untied, local signal model* $\boldsymbol{U}_{[K]}^\ell$: each layer models the distribution of input tokens $\boldsymbol{Z}^\ell$, enabling the linearization of nonlinear structures in the input distribution $\boldsymbol{X}$ at a global scale over the course of the application of $f$.

The above unrolled optimization framework admits a variety of design choices to realize the layers $f^\ell$ that incrementally optimize (2). CRATE employs a two-stage alternating minimization approach with a strong conceptual basis [51], which we summarize here and describe in detail below:

1. First, the distribution of tokens $\boldsymbol{Z}^\ell$ is *compressed* against the local signal model $\boldsymbol{U}_{[K]}^\ell$ by an approximate gradient step on $R^c(\boldsymbol{Z} \mid \boldsymbol{U}_{[K]}^\ell)$ to create an intermediate representation $\boldsymbol{Z}^{\ell+1/2}$;

2. Second, this intermediate representation is *sparsely encoded* using a learnable dictionary $\boldsymbol{D}^\ell$ to generate the next layer representation $\boldsymbol{Z}^{\ell+1}$.

Experiments demonstrate that even after supervised training, CRATE achieves its design goals for representation learning articulated above [51].

**Compression operator: Multi-Head Subspace Self-Attention (MSSA).** Given local models $\boldsymbol{U}_{[K]}^\ell$, to form the incremental transformation $f^\ell$ optimizing (2) at layer $\ell$, CRATE first compresses the token set $\boldsymbol{Z}^\ell$ against the subspaces by minimizing the coding rate $R^c(\,\cdot\,\mid \boldsymbol{U}_{[K]}^\ell)$. As Yu et al. [51] show, doing this minimization locally by performing a step of gradient descent on $R^c(\,\cdot\,\mid \boldsymbol{U}_{[K]}^\ell)$ leads to the so-called multi-head subspace self-attention (MSSA) operation, defined as

$$\texttt{MSSA}(\boldsymbol{Z} \mid \boldsymbol{U}_{[K]}) \doteq \frac{p}{(N+1)\varepsilon^2} [\boldsymbol{U}_1, \ldots, \boldsymbol{U}_K] \begin{bmatrix} (\boldsymbol{U}_1^*\boldsymbol{Z})\,\mathrm{softmax}((\boldsymbol{U}_1^*\boldsymbol{Z})^*(\boldsymbol{U}_1^*\boldsymbol{Z})) \\ \vdots \\ (\boldsymbol{U}_K^*\boldsymbol{Z})\,\mathrm{softmax}((\boldsymbol{U}_K^*\boldsymbol{Z})^*(\boldsymbol{U}_K^*\boldsymbol{Z})) \end{bmatrix}, \quad (5)$$

and the subsequent intermediate representation

$$\boldsymbol{Z}^{\ell+1/2} = \boldsymbol{Z}^\ell - \kappa \nabla_{\boldsymbol{Z}} R^c(\boldsymbol{Z}^\ell \mid \boldsymbol{U}_{[K]}) \approx \left(1 - \kappa \cdot \frac{p}{(N+1)\varepsilon^2}\right) \boldsymbol{Z}^\ell + \kappa \cdot \frac{p}{(N+1)\varepsilon^2} \cdot \texttt{MSSA}(\boldsymbol{Z}^\ell \mid \boldsymbol{U}_{[K]}), \quad (6)$$

where $\kappa > 0$ is a learning rate hyperparameter. This block bears a striking resemblance to the ViT's multi-head self-attention block, with a crucial difference: the usual query, key, and value projection matrices within a single head are here all identical, and determined by our local model for the distribution of the input tokens. We will demonstrate via careful ablation studies that this distinction is crucial for the emergence of useful visual representations in CRATE.

**Sparsification operator: Iterative Shrinkage-Thresholding Algorithm (ISTA).** The remaining term to optimize in (2) is the difference of the global coding rate $R(\boldsymbol{Z})$ and the $\ell^0$ norm of the tokens, which together encourage the representations to be both sparse and non-collapsed. Yu et al. [51] show that local minimization of this objective in a neighborhood of the intermediate representations $\boldsymbol{Z}^{\ell+1/2}$ is approximately achieved by a LASSO problem with respect to a sparsifying orthogonal dictionary $\boldsymbol{D}^\ell$. Taking an iterative step towards solving this LASSO problem gives the iterative shrinkage-thresholding algorithm (ISTA) block [47, 51]:

$$\boldsymbol{Z}^{\ell+1} = f^\ell(\boldsymbol{Z}^\ell) = \mathrm{ReLU}(\boldsymbol{Z}^{\ell+1/2} + \eta \boldsymbol{D}^{\ell*}(\boldsymbol{Z}^{\ell+1/2} - \boldsymbol{D}^\ell \boldsymbol{Z}^{\ell+1/2}) - \eta\lambda\mathbf{1}) \doteq \texttt{ISTA}(\boldsymbol{Z}^{\ell+1/2} \mid \boldsymbol{D}^\ell). \quad (7)$$

Here, $\eta > 0$ is a step size, and $\lambda > 0$ is the sparsification regularizer. The ReLU nonlinearity appearing in this block arises from an additional nonnegativity constraint on the representations in the LASSO program, motivated by the goal of better separating distinct modes of variability in the token distribution [17]. The ISTA block is reminiscent of the MLP block in the ViT, but with a relocated skip connection.

**The overall CRATE architecture.** Combining the MSSA and the ISTA block, as above, together with a suitable choice of hyperparameters, we arrive at the definition of a single CRATE layer:

$$\boldsymbol{Z}^{\ell+1/2} \doteq \boldsymbol{Z}^\ell + \texttt{MSSA}(\boldsymbol{Z}^\ell \mid \boldsymbol{U}_{[K]}^\ell), \qquad f^\ell(\boldsymbol{Z}^\ell) = \boldsymbol{Z}^{\ell+1} \doteq \texttt{ISTA}(\boldsymbol{Z}^{\ell+1/2} \mid \boldsymbol{D}^\ell). \quad (8)$$

These layers are composed to obtain the representation $f$, as in (4). We visualize the CRATE architecture in Figure 2. Full pseudocode (both mathematical and PyTorch-style) is given in Appendix A.

**The forward and backward pass of CRATE.** The above conceptual framework separates the role of *forward "optimization,"* where each layer incrementally transforms its input towards a compact and structured representation via compression and sparsification of the token representations using the local signal models $\boldsymbol{U}_{[K]}^\ell$ and sparsifying dictionaries $\boldsymbol{D}^\ell$ at each layer, and *backward "learning,"* where the local signal models and sparsifying dictionaries are learned from supervised (as in our experiments) or self-supervised training via back propagation to capture structures in the data. We believe that such mathematically clear designs of CRATE play a key role in the emergence of semantically meaningful properties in the final learned models, as we will soon see.

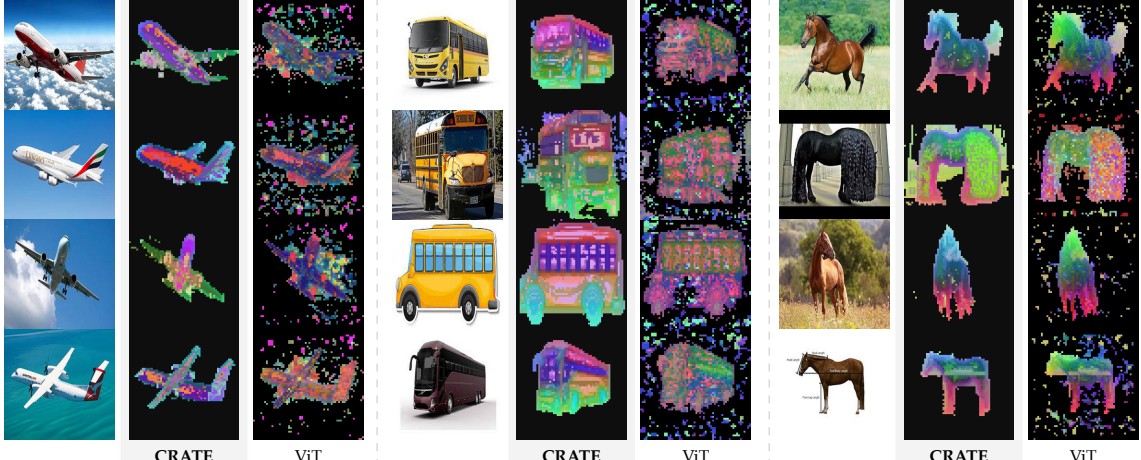

**Figure 3: Visualization of PCA components.** We compute the PCA of the patch-wise representations of each column and visualize the first 3 components for the foreground object. Each component is matched to a different RGB channel and the background is removed by thresholding the first PCA component of the full image. The representations of CRATE are better aligned, and with less spurious correlations, to texture and shape components of the input than those of ViT. See the pipeline in Appendix B.2 for more details.

## 2.2. Training CRATE with Supervised Learning

As described in previous subsection, given the *local signal models* $(\boldsymbol{U}_{[K]}^{\ell})_{\ell=1}^{L}$ and *sparsifying dictionaries* $(\boldsymbol{D}^{\ell})_{\ell=1}^{L}$, each layer of CRATE is designed to optimize the sparse rate reduction objective (2). To enable more effective compression and sparsification, the parameters of local signal models need to be identified. Previous work [51] proposes to learn the parameters $(\boldsymbol{U}_{[K]}^{\ell}, \boldsymbol{D}^{\ell})_{\ell=1}^{L}$ from data, specifically in a supervised manner by solving the following classification problem:

$$\min_{\boldsymbol{W}, f} \sum_i \ell_{\text{CE}}(\boldsymbol{W}\boldsymbol{z}_{i,[\text{CLS}]}^{L+1}, y_i) \quad \text{where} \quad \left[\boldsymbol{z}_{i,[\text{CLS}]}^{L+1}, \boldsymbol{z}_{i,1}^{L+1}, \dots, \boldsymbol{z}_{i,N}^{L+1}\right] = f(\boldsymbol{X}_i), \tag{9}$$

where $(\boldsymbol{X}_i, y_i)$ is the $i^{\text{th}}$ training (image, label) pair, $\boldsymbol{W} \in \mathbb{R}^{d \times C}$ maps the [CLS] token to a vector of logits, $C$ is the number of classes, and $\ell_{\text{CE}}(\cdot, \cdot)$ denotes the softmax cross-entropy loss.[1]

# 3. Measuring Emerging Properties in CRATE

We now study the emergent segmentation properties in supervised CRATE both qualitatively and quantitatively. As demonstrated in previous work [6], segmentation within the ViT [11] emerges only when applying DINO, a very specialized self-supervised learning method [6]. In particular, *a vanilla ViT trained on supervised classification does not develop the ability to perform segmentation.* In contrast, as we demonstrate both qualitatively and quantitatively in Section 3 and Section 4, *segmentation properties emerge in CRATE even when using standard supervised classification training.*

Our empirical results demonstrate that self-supervised learning, as well as the specialized design options in DINO [6] (e.g., momentum encoder, student and teacher networks, self-distillation, etc.) are not necessary for the emergence of segmentation. We train all models (CRATE and ViT) with the same number of data and iterations, as well as optimizers, to ensure experiments and ablations provide a fair comparison—precise details are provided in Appendix C.1.

## 3.1. Qualitative Measurements

**Visualizing self-attention maps.** To qualitatively measure the emergence phenomenon, we adopt the attention map approach based on the [CLS] token, which has been widely used as a way to interpret and visualize transformer-like architectures [1, 6]. Indeed, we use the same methodology as [1, 6], noting that in CRATE the query-key-value matrices are all the same; a more formal accounting is deferred to Appendix B.1. The visualization results of self-attention maps are summarized in

---

[1]This is similar to the supervised ViT training used in Dosovitskiy et al. [11].

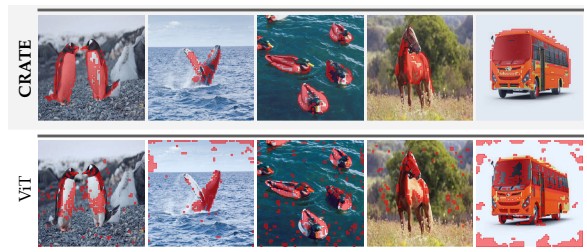

| Model | Train | mIoU |
|---|---|---|
| CRATE-S/8 | Supervised | 23.9 |
| CRATE-B/8 | Supervised | 23.6 |
| ViT-S/8 | Supervised | 14.1 |
| ViT-B/8 | Supervised | 19.2 |
| ViT-S/8 | DINO | 27.0 |
| ViT-B/8 | DINO | 27.3 |

(**a**) Visualization of coarse semantic segmentation.      (**b**) mIoU evaluation.

**Figure 4: Coarse semantic segmentation via self-attention map.** (*a*) We visualize the segmentation masks for both CRATE and the supervised ViT. We select the attention head with the best segmentation performance for CRATE and ViT separately. (*b*) We quantitatively evaluate the coarse segmentation mask by evaluating the mIoU score on the validation set of PASCAL VOC12 [13]. Overall, CRATE demonstrates superior segmentation performance to the supervised ViT both qualitatively (e.g., in (a), where the segmentation maps are much cleaner and outline the desired object), and quantitatively (e.g., in (b)).

Figure 1 and Figure 7. We observe that the self-attention maps of the CRATE model correspond to semantic regions in the input image. Our results suggest that the CRATE model encodes a clear semantic segmentation of each image in the network's internal representations, which is similar to the self-supervised method DINO [6]. In contrast, as shown in Figure 14 in the Appendices, the vanilla ViT trained on supervised classification does not exhibit similar segmentation properties.

**PCA visualization for patch-wise representation.** Following previous work [2, 35] on visualizing the learned patch-wise deep features of image, we study the principal component analysis (PCA) on the deep token representations of CRATE and ViT models. Again, we use the same methodology as the previous studies [2, 35], and a more full accounting of the method is deferred to Appendix B.2. We summarize the PCA visualization results of supervised CRATE in Figure 3. Without segmentation supervision, CRATE is able to capture the boundary of the object in the image. Moreover, the principal components demonstrate feature alignment between tokens corresponding to similar parts of the object; for example, the red channel corresponds to the horse's leg. On the other hand, the PCA visualization of the supervised ViT model is considerably less structured. We also provide more PCA visualization results in Figure 9.

## 3.2. Quantitative Measurements

Besides qualitatively assessing segmentation properties through visualization, we also quantitatively evaluate the emergent segmentation property of CRATE using existing segmentation and object detection techniques [6, 46]. Both methods apply the internal deep representations of transformers, such as the previously discussed self-attention maps, to produce segmentation masks without further training on special annotations (e.g., object boxes, masks, etc.).

**Coarse segmentation via self-attention map.** As shown in Figure 1, CRATE explicitly captures the object-level semantics with clear boundaries. To quantitatively measure the quality of the induced segmentation, we utilize the raw self-attention maps discussed earlier to generate segmentation masks. Then, we evaluate the standard mIoU (mean intersection over union) score [28] by comparing the generated segmentation masks against ground truth masks. This approach has been used in previous work on evaluating the segmentation performance of the self-attention maps [6]. A more detailed accounting of the methodology is found in Appendix B.3. The results are summarized in Figure 4. CRATE largely outperforms ViT both visually and in terms of mIoU, which suggests that the internal representations in CRATE are much more effective for producing segmentation masks.

**Object detection and fine-grained segmentation.** To further validate and evaluate the rich semantic information captured by CRATE, we employ MaskCut [46], a recent effective approach for object detection and segmentation that does not require human annotations. As usual, we provide a more detailed methodological description in Appendix B.4. This procedure allows us to extract more fine-grained segmentation from an image based on the token representations learned in CRATE. We visualize the fine-grained segmentations produced by MaskCut in Figure 5 and compare the segmentation and detection performance in Table 1. Based on these results, we observe that MaskCut with supervised ViT features completely fails to produce segmentation masks in certain cases, for example, the first image in Figure 5 and the ViT-S/8 row in Table 1. Compared to ViT, CRATE provides better internal representation tokens for both segmentation and detection.

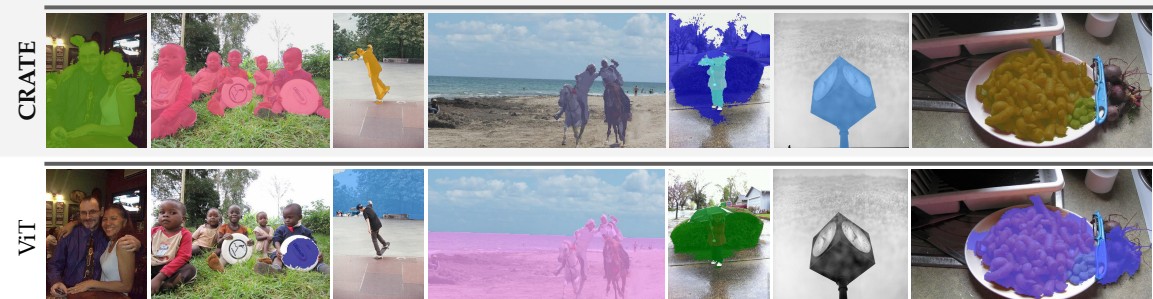

**Figure 5: Visualization of on COCO val2017 [27] with MaskCut.** (*Top Row*) Supervised **CRATE** architecture clearly detects the major objects in the image. (*Bottom Row*) Supervised ViT sometimes fails to detect the major objects in the image (columns 2, 3, 4).

| Model | Train | Detection | | | Segmentation | | |
|---|---|---|---|---|---|---|---|
| | | $AP_{50}$ | $AP_{75}$ | AP | $AP_{50}$ | $AP_{75}$ | AP |
| **CRATE**-S/8 | Supervised | 2.9 | 1.0 | 1.1 | 1.8 | 0.7 | 0.8 |
| **CRATE**-B/8 | Supervised | 2.9 | 1.0 | 1.3 | 2.2 | 0.7 | 1.0 |
| ViT-S/8 | Supervised | 0.1 | 0.1 | 0.0 | 0.0 | 0.0 | 0.0 |
| ViT-B/8 | Supervised | 0.8 | 0.2 | 0.4 | 0.7 | 0.5 | 0.4 |
| ViT-S/8 | DINO | 5.0 | 2.0 | 2.4 | 4.0 | 1.3 | 1.7 |
| ViT-B/8 | DINO | 5.1 | 2.3 | 2.5 | 4.1 | 1.3 | 1.8 |

**Table 1: Object detection and fine-grained segmentation via MaskCut on COCO val2017 [27].** We consider models with different scales and evaluate the average precision measured by COCO's official evaluation metric. The first four models are pre-trained with image classification tasks under label supervision; the bottom two models are pre-trained via the DINO self-supervised technique [6]. **CRATE** conclusively performs better than the ViT at detection and segmentation metrics when both are trained using supervised classification.

## 4. White-Box Empowered Analysis of Segmentation in **CRATE**

In this section, we delve into the segmentation properties of **CRATE** using analysis powered by our white-box perspective. To start with, we analyze the internal token representations from different layers of **CRATE** and study the power of the network segmentation as a function of the layer depth. We then perform an ablation study on various architectural configurations of **CRATE** to isolate the essential components for developing segmentation properties. Finally, we investigate how to identify the "semantic" meaning of certain subspaces and extract more fine-grained information from **CRATE**. We use the **CRATE**-B/8 and ViT-B/8 as the default models for evaluation in this section.

**Role of depth in **CRATE**.** Each layer of **CRATE** is designed for the same conceptual purpose: to optimize the sparse rate reduction and transform the token distribution to compact and structured forms (Section 2). Given that the emergence of semantic segmentation in **CRATE** is analogous to the *clustering of tokens belonging to similar semantic categories in the representation $Z$*, we therefore expect the segmentation performance of **CRATE** to improve with increasing depth. To test this, we utilize the MaskCut pipeline (described in Section 3.2) to quantitatively evaluate the segmentation performance of the internal representations across different layers. Meanwhile, we apply the PCA visualization (described in Section 3.1) for understanding how segmentation emerges with respect to depth. Compared to the results in Figure 3, a minor difference in our visualization is that we show the first four principal components in Figure 6 and do not filter out background tokens.

The results are summarized in Figure 6. We observe that the segmentation score improves when using representations from deeper layers, which aligns well with the incremental optimization design of **CRATE**. In contrast, even though the performance of ViT-B/8 slightly improves in later layers, its segmentation scores are significantly lower than those of **CRATE** (c.f. failures in Figure 5, bottom row). The PCA results are presented in Figure 6 (*Right*). We observe that representations extracted from deeper layers of **CRATE** increasingly focus on the foreground object and are able to capture texture-level details. Figure 9 in the Appendix has more PCA visualization results.

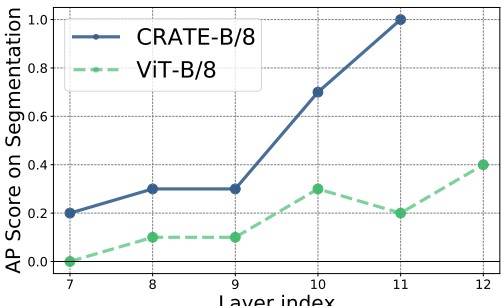 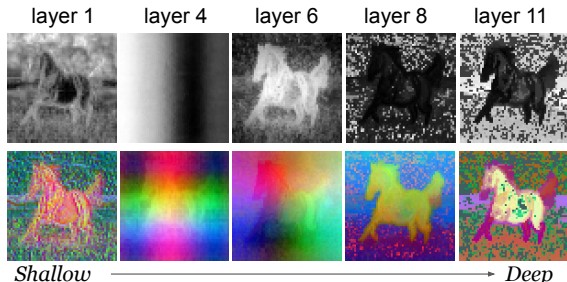

**Figure 6: Effect of depth for segmentation in supervised CRATE.** (*Left*) Layer-wise segmentation performance of **CRATE** and ViT via MaskCut pipeline on COCO val2017 (Higher AP score means better segmentation performance). (*Right*) We follow the implementation in Amir et al. [2]: we first apply PCA on patch-wise features. Then, for the gray figure, we visualize the 1st components, and for the colored figure, we visualize the 2nd, 3rd and 4th components, which correspond to the RGB color channels. See more results in Figure 9.

| Model | Attention | Nonlinearity | COCO Detection | | | VOC Seg. |
|---|---|---|---|---|---|---|
| | | | AP$_{50}$ | AP$_{75}$ | AP | mIoU |
| **CRATE** | MSSA | ISTA | 2.1 | 0.7 | 0.8 | 23.9 |
| **CRATE-MLP** | MSSA | MLP | 0.2 | 0.2 | 0.2 | 22.0 |
| **CRATE-MHSA** | MHSA | ISTA | 0.1 | 0.1 | 0.0 | 18.4 |
| ViT | MHSA | MLP | 0.1 | 0.1 | 0.0 | 14.1 |

**Table 2: Ablation study of different CRATE variants.** We use the `Small-Patch8` (`S-8`) model configuration across all experiments in this table.

**Ablation study of architecture in CRATE.** Both the attention block (MSSA) and the MLP block (ISTA) in CRATE are different from the ones in the ViT. In order to understand the effect of each component for emerging segmentation properties of CRATE, we study three different variants of CRATE: CRATE, CRATE-MHSA, CRATE-MLP, where we denote the attention block and MLP block in ViT as MHSA and MLP respectively. We summarize different model architectures in Table 2.

For all models in Table 2, we apply the same pre-training setup on the ImageNet-21k dataset. We then apply the coarse segmentation evaluation (Section 3.2) and MaskCut evaluation (Section 3.2) to quantitatively compare the performance of different models. As shown in Table 2, CRATE significantly outperforms other model architectures across all tasks. Interestingly, we find that the coarse segmentation performance (i.e., VOC Seg) of the ViT can be significantly improved by simply replacing the MHSA in ViT with the MSSA in CRATE, despite the architectural differences between MHSA and MSSA being small. This demonstrates the effectiveness of the white-box design.

**Identifying semantic properties of attention heads.** As shown in Figure 1, the self-attention map between the [CLS] token and patch tokens contains clear segmentation masks. We are interested in capturing the semantic meaning of certain attention *heads*; this is an important task for interpretability, and is already studied for language transformers [34]. Intuitively, each head captures certain features of the data. Given a CRATE model, we first forward an input image (e.g. a horse image as in Figure 7) and select four attention heads which seem to have semantic meaning by manual inspection. After identifying the attention heads, we visualize the self-attention map of these heads on other input images. We visualize the results in Figure 7. Interestingly, we find that each of the selected attention heads captures a different part of the object, and even a different semantic meaning. For example, the attention head displayed in the first column of Figure 7 captures the legs of different animals, and the attention head displayed in the last column captures the ears and head. This parsing of the visual input into a part-whole hierarchy has been a fundamental goal of learning-based recognition architectures since deformable part models [14, 15] and capsule networks [20, 40]—strikingly, it emerges from the white-box design of CRATE within our simple supervised training setup.[2]

---

[2]In this connection, we note that Hinton [19] recently surmised that the query, key, and value projections in the transformer should be made equal for this reason—the design of CRATE and Figure 7 confirm this.

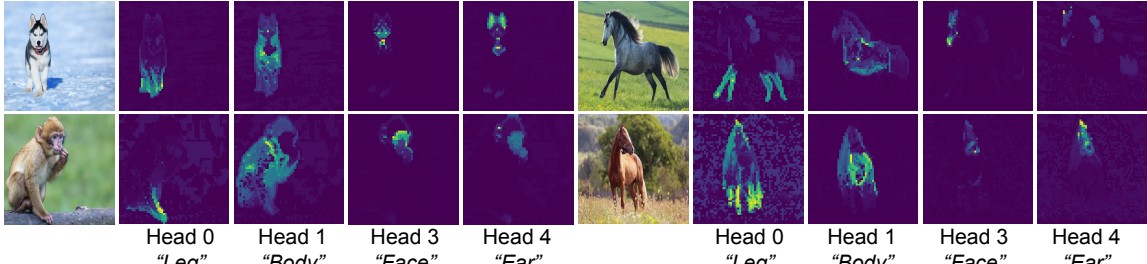

| Head 0 | Head 1 | Head 3 | Head 4 | | Head 0 | Head 1 | Head 3 | Head 4 |
| *"Leg"* | *"Body"* | *"Face"* | *"Ear"* | | *"Leg"* | *"Body"* | *"Face"* | *"Ear"* |

**Figure 7: Visualization of semantic heads.** We forward a mini-batch of images through a supervised CRATE and examine the attention maps from all the heads in the penultimate layer. We visualize a selection of attention heads to show that certain heads convey specific semantic meaning, i.e. *head 0 ↔ "Legs", head 1 ↔ "Body", head 3 ↔ "Face", head 4 ↔ "Ear"*. See more in Figure 15
.

# 5. Related Work

**Visual attention and emergence of segmentation.** The concept of attention has become increasingly significant in intelligence, evolving from early computational models [21, 23, 41] to modern neural networks [11, 44]. In deep learning, the self-attention mechanism has been widely employed in processing visual data [11] with state-of-the-art performance on various visual tasks [6, 18, 35].

DINO [6] demonstrated that attention maps generated by *self-supervised* Vision Transformers (ViT)[11] can implicitly perform semantic segmentation of images. This emergence of segmentation capability has been corroborated by subsequent *self-supervised* learning studies [6, 18, 35]. Capitalizing on these findings, recent segmentation methodologies [2, 22, 46] have harnessed these emergent segmentations to attain *state-of-the-art* results. Nonetheless, there is a consensus, as highlighted in studies like Caron et al. [6], suggesting that such segmentation capability would not manifest in a supervised ViT. A key motivation and contribution of our research is to show that transformer-like architectures, as in CRATE, can exhibit this ability even with supervised training.

**White-box models.** In data analysis, there has continually been significant interest in developing interpretable and structured representations of the dataset. The earliest manifestations of such interest were in sparse coding via dictionary learning [47], which are white-box models that transform the (approximately linear) data into human-interpretable standard forms (highly sparse vectors). The advent of deep learning has not changed this desire much, and indeed attempts have been made to marry the power of deep learning with the interpretability of white-box models. Such attempts include scattering networks [5], convolutional sparse coding networks [36], and the sparse manifold transform [9]. Another line of work constructs deep networks from unrolled optimization [7, 43, 50, 51]. Such models are fully interpretable, yet only recently have they demonstrated competitive performance with black-box alternatives such as ViT at ImageNet scale [51]. This work builds on one such powerful white-box model, CRATE [51], and demonstrates more of its capabilities, while serving as an example for the fine-grained analysis made possible by white-box models.

# 6. Discussions and Future Work

In this study, we demonstrated that when employing the white-box model CRATE as a foundational architecture in place of the ViT, there is a natural emergence of segmentation masks even while using a straightforward supervised training approach. Our empirical findings underscore the importance of principled architecture design for developing better vision foundation models. As simpler models are more interpretable and easier to analyze, we are optimistic that the insights derived from white-box transformers in this work will contribute to a deeper empirical and theoretical understanding of the segmentation phenomenon. Furthermore, our findings suggest that white-box design principles hold promise in offering concrete guidelines for developing enhanced vision foundation models. Two compelling directions for further research would be investigating how to better engineer white-box models such as CRATE to match the performance of self-supervised learning methods (such as DINO), and expanding the range of tasks for which white-box models are practically useful.

**Acknowledgements.** We thank Xudong Wang and Baifeng Shi for valuable discussions on segmentation properties in vision transformers.

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

# Appendix

## A. CRATE Implementation

In this section, we provide the details on our implementation of CRATE, both at a higher level for use in mathematical analysis, and at a code-based level for use in reference implementations. While we used the same implementation as in Yu et al. [51], we provide the details here for completeness.

### A.1. Forward-Pass Algorithm

We provide the details on the forward pass of CRATE in Algorithm 1.

---

**Algorithm 1** CRATE Forward Pass.

---

**Hyperparameter:** Number of layers $L$, feature dimension $d$, subspace dimension $p$, image dimension $(C, H, W)$, patch dimension $(P_H, P_W)$, sparsification regularizer $\lambda > 0$, quantization error $\varepsilon$, learning rate $\eta > 0$.

**Parameter:** Patch projection matrix $\boldsymbol{W} \in \mathbb{R}^{d \times D}$.  $\qquad \qquad \qquad \triangleright D \doteq P_H P_W$.
**Parameter:** Class token $\boldsymbol{z}^0_{\texttt{[CLS]}} \in \mathbb{R}^d$.
**Parameter:** Positional encoding $\boldsymbol{E}_{\text{pos}} \in \mathbb{R}^{d \times (N+1)}$.  $\qquad \qquad \triangleright N \doteq \frac{H}{P_H} \cdot \frac{W}{P_W}$.
**Parameter:** Local signal models $(\boldsymbol{U}^\ell_{[K]})^L_{\ell=1}$ where each $\boldsymbol{U}^\ell_{[K]} = (\boldsymbol{U}^\ell_1, \ldots, \boldsymbol{U}^\ell_K)$ and each $\boldsymbol{U}^\ell_k \in \mathbb{R}^{d \times p}$.
**Parameter:** Sparsifying dictionaries $(\boldsymbol{D}^\ell)^L_{\ell=1}$ where each $\boldsymbol{D}^\ell \in \mathbb{R}^{d \times d}$.
**Parameter:** Sundry LAYERNORM parameters.

1: **function** MSSA($\boldsymbol{Z} \in \mathbb{R}^{d \times (N+1)} \mid \boldsymbol{U}_{[K]} \in \mathbb{R}^{K \times d \times p}$)
2: $\qquad$ **return** $\dfrac{p}{(N+1)\varepsilon^2} \displaystyle\sum_{k=1}^K \boldsymbol{U}_k(\boldsymbol{U}^*_k \boldsymbol{Z}) \operatorname{softmax}((\boldsymbol{U}^*_k \boldsymbol{Z})^*(\boldsymbol{U}^*_k \boldsymbol{Z}))$  $\qquad \triangleright$ Eq. (5)
3: **end function**

4: **function** ISTA($\boldsymbol{Z} \in \mathbb{R}^{d \times (N+1)} \mid \boldsymbol{D} \times \mathbb{R}^{d \times d}$)
5: $\qquad$ **return** $\operatorname{ReLU}(\boldsymbol{Z} + \eta \boldsymbol{D}^*(\boldsymbol{Z} - \boldsymbol{D}\boldsymbol{Z}) - \eta\lambda\mathbf{1})$  $\qquad \qquad \qquad \triangleright$ Eq. (7)
6: **end function**

7: **function** CRATEFORWARDPASS(IMG $\in \mathbb{R}^{C \times H \times W}$)
8: $\qquad \boldsymbol{X} \doteq [\boldsymbol{x}_1, \ldots, \boldsymbol{x}_N] \leftarrow$ PATCHIFY(IMG)  $\qquad \triangleright \boldsymbol{X} \in \mathbb{R}^{D \times N}$ and each $\boldsymbol{x}_i \in \mathbb{R}^D$.

9: $\qquad$ # $f^0$ Operator
10: $\qquad [\boldsymbol{z}^1_1, \ldots, \boldsymbol{z}^1_N] \leftarrow \boldsymbol{W}\boldsymbol{X}$  $\qquad \qquad \qquad \qquad \qquad \qquad \triangleright \boldsymbol{z}^1_i \in \mathbb{R}^d$.
11: $\qquad \boldsymbol{Z}^1 \leftarrow [\boldsymbol{z}^1_{\texttt{[CLS]}}, \boldsymbol{z}^1_1, \ldots, \boldsymbol{z}^1_N] + \boldsymbol{E}_{\text{pos}}$  $\qquad \qquad \qquad \triangleright \boldsymbol{Z}^1 \in \mathbb{R}^{d \times (N+1)}$.

12: $\qquad$ # $f^\ell$ Operators
13: $\qquad$ **for** $\ell \in \{1, \ldots, L\}$ **do**
14: $\qquad \qquad \boldsymbol{Z}^\ell_n \leftarrow$ LAYERNORM($\boldsymbol{Z}^\ell$)  $\qquad \qquad \qquad \qquad \triangleright \boldsymbol{Z}^\ell_n \in \mathbb{R}^{d \times (N+1)}$
15: $\qquad \qquad \boldsymbol{Z}^{\ell+1/2} \leftarrow \boldsymbol{Z}^\ell_n +$ MSSA($\boldsymbol{Z}^\ell_n \mid \boldsymbol{U}^\ell_{[K]}$)  $\qquad \triangleright \boldsymbol{Z}^{\ell+1/2} \in \mathbb{R}^{d \times (N+1)}$
16: $\qquad \qquad \boldsymbol{Z}^{\ell+1/2}_n \leftarrow$ LAYERNORM($\boldsymbol{Z}^{\ell+1/2}$)  $\qquad \qquad \triangleright \boldsymbol{Z}^{\ell+1/2}_n \in \mathbb{R}^{d \times (N+1)}$
17: $\qquad \qquad \boldsymbol{Z}^{\ell+1} \leftarrow$ ISTA($\boldsymbol{Z}^{\ell+1/2}_n \mid \boldsymbol{D}^\ell$)  $\qquad \qquad \quad \triangleright \boldsymbol{Z}^{\ell+1} \in \mathbb{R}^{d \times (N+1)}$
18: $\qquad$ **end for**

19: $\qquad$ **return** $\boldsymbol{Z} \leftarrow \boldsymbol{Z}^{L+1}$
20: **end function**

---

## A.2. PyTorch-Like Code for Forward Pass

Similar to the previous subsection, we provide the pseudocode for the MSSA block and ISTA block in Algorithm 2, and then present the pseudocode for the forward pass of CRATE in Algorithm 3.

---

**Algorithm 2** PyTorch-Like Code for MSSA and ISTA Forward Passes

---

```python
class ISTA:
    # initialization
    def __init__(self, dim, hidden_dim, dropout = 0., step_size=0.1, lambd=0.1):
        self.weight = Parameter(Tensor(dim, dim))
        init.kaiming_uniform_(self.weight)
        self.step_size = step_size
        self.lambd = lambd
    # forward pass
    def forward(self, x):
        x1 = linear(x, self.weight, bias=None)
        grad_1 = linear(x1, self.weight.t(), bias=None)
        grad_2 = linear(x, self.weight.t(), bias=None)
        grad_update = self.step_size * (grad_2 - grad_1) - self.step_size * self.lambd
        output = relu(x + grad_update)
        return output
class MSSA:
    # initialization
    def __init__(self, dim, heads = 8, dim_head = 64, dropout = 0.):
        inner_dim = dim_head * heads
        project_out = not (heads == 1 and dim_head == dim)
        self.heads = heads
        self.scale = dim_head ** -0.5
        self.attend = Softmax(dim = -1)
        self.dropout = Dropout(dropout)
        self.qkv = Linear(dim, inner_dim, bias=False)
        self.to_out = Sequential(Linear(inner_dim, dim), Dropout(dropout)) if project_out
    else nn.Identity()
    # forward pass
    def forward(self, x):
        w = rearrange(self.qkv(x), 'b n (h d) -> b h n d', h = self.heads)
        dots = matmul(w, w.transpose(-1, -2)) * self.scale
        attn = self.attend(dots)
        attn = self.dropout(attn)
        out = matmul(attn, w)
        out = rearrange(out, 'b h n d -> b n (h d)')
        return self.to_out(out)
```

---

**Algorithm 3** PyTorch-Like Code for CRATE Forward Pass

---

```python
class CRATE:
    # initialization
    def __init__(self, dim, depth, heads, dim_head, mlp_dim, dropout = 0.):
        # define layers
        self.layers = []
        self.depth = depth
        for _ in range(depth):
            self.layers.extend([LayerNorm(dim), MSSA(dim, heads, dim_head, dropout)])
            self.layers.extend([LayerNorm(dim), ISTA(dim, mlp_dim, dropout)])
    # forward pass
    def forward(self, x):
        for ln1, attn, ln2, ff in self.layers:
            x_ = attn(ln1(x)) + ln1(x)
            x = ff(ln2(x_))
        return x
```

---

# B. Detailed Experimental Methodology

In this section we formally describe each of the methods used to evaluate the segmentation property of CRATE in Section 3 and Section 4, especially compared to DINO and supervised ViT. This section repeats experimental methodologies covered less formally in other works; we strive to rigorously define the experimental methodologies in this section.

## B.1. Visualizing Attention Maps

We recapitulate the method to visualize attention maps in Abnar and Zuidema [1] and Caron et al. [6], at first specializing their use to instances of the CRATE model before generalizing to the ViT.

For the $k^{\text{th}}$ head at the $\ell^{\text{th}}$ layer of CRATE, we compute the *self-attention matrix* $\boldsymbol{A}_k^\ell \in \mathbb{R}^N$ defined as follows:

$$\boldsymbol{A}_k^\ell = \begin{bmatrix} A_{k,1}^\ell \\ \vdots \\ A_{k,N}^\ell \end{bmatrix} \in \mathbb{R}^N, \quad \text{where} \quad A_{k,i}^\ell = \frac{\exp(\langle \boldsymbol{U}_k^{\ell*}\boldsymbol{z}_i^\ell, \boldsymbol{U}_k^{\ell*}\boldsymbol{z}_{[\text{CLS}]}^\ell \rangle)}{\sum_{j=1}^N \exp(\langle \boldsymbol{U}_k^{\ell*}\boldsymbol{z}_j^\ell, \boldsymbol{U}_k^{\ell*}\boldsymbol{z}_{[\text{CLS}]}^\ell \rangle)}. \tag{10}$$

We then reshape the attention matrix $\boldsymbol{A}_k^\ell$ into a $\sqrt{N} \times \sqrt{N}$ matrix and visualize the heatmap as shown in Figure 1. For example, the $i^{\text{th}}$ row and the $j^{\text{th}}$ column element of the heatmap in Figure 1 corresponds to the $m^{\text{th}}$ component of $\boldsymbol{A}_k^\ell$ if $m = (i-1) \cdot \sqrt{N} + j$. In Figure 1, we select one attention head of CRATE and visualize the attention matrix $\boldsymbol{A}_k^\ell$ for each image.

For the ViT, the entire methodology remains the same, except that the attention map is defined in the following reasonable way:

$$\boldsymbol{A}_k^\ell = \begin{bmatrix} A_{k,1}^\ell \\ \vdots \\ A_{k,N}^\ell \end{bmatrix} \in \mathbb{R}^N, \quad \text{where} \quad A_{k,i}^\ell = \frac{\exp(\langle \boldsymbol{K}_k^{\ell*}\boldsymbol{z}_i^\ell, \boldsymbol{Q}_k^{\ell*}\boldsymbol{z}_{[\text{CLS}]}^\ell \rangle)}{\sum_{j=1}^N \exp(\langle \boldsymbol{K}_k^{\ell*}\boldsymbol{z}_j^\ell, \boldsymbol{Q}_k^{\ell*}\boldsymbol{z}_{[\text{CLS}]}^\ell \rangle)}. \tag{11}$$

where the "query" and "key" parameters of the standard transformer at head $k$ and layer $\ell$ are denoted $\boldsymbol{K}_k^\ell$ and $\boldsymbol{Q}_k^\ell$ respectively.

## B.2. PCA Visualizations

As in the previous subsection, we recapitulate the method to visualize the patch representations using PCA from Amir et al. [2] and Oquab et al. [35]. As before we specialize their use to instances of the CRATE model before generalizing to the ViT.

We first select $J$ images that belong to the same class, $\{\boldsymbol{X}_j\}_{j=1}^J$, and extract the token representations for each image at layer $\ell$, i.e., $[\boldsymbol{z}_{j,[\text{CLS}]}^\ell, \boldsymbol{z}_{j,1}^\ell, \ldots, \boldsymbol{z}_{j,N}^\ell]$ for $j \in [J]$. In particular, $\boldsymbol{z}_{j,i}^\ell$ represents the $i^{\text{th}}$ token representation at the $\ell^{\text{th}}$ layer for the $j^{\text{th}}$ image. We then compute the first PCA components of $\widehat{\boldsymbol{Z}}^\ell = \{\widehat{\boldsymbol{z}}_{1,1}^\ell, \ldots, \widehat{\boldsymbol{z}}_{1,N}^\ell, \ldots, \widehat{\boldsymbol{z}}_{J,1}^\ell, \ldots, \widehat{\boldsymbol{z}}_{J,N}^\ell\}$, and use $\widehat{\boldsymbol{z}}_{j,i}^\ell$ to denote the aggregated token representation for the $i$-th token of $\boldsymbol{X}_j$, i.e., $\widehat{\boldsymbol{z}}_{j,i}^\ell = [(\boldsymbol{U}_1^*\widehat{\boldsymbol{z}}_{j,i}^\ell)^\top, \ldots, (\boldsymbol{U}_K^*\widehat{\boldsymbol{z}}_{j,i}^\ell)^\top]^\top \in \mathbb{R}^{(p \cdot K) \times 1}$. We denote the first eigenvector of the matrix $\widehat{\boldsymbol{Z}}^*\widehat{\boldsymbol{Z}}$ by $\boldsymbol{u}_0$ and compute the projection values as $\{\sigma_\lambda(\langle \boldsymbol{u}_0, \boldsymbol{z}_{j,i}^\ell \rangle)\}_{i,j}$, where $\sigma_\lambda(x) = \begin{cases} x, & |x| \geq \lambda \\ 0, & |x| < \lambda \end{cases}$ is the hard-thresholding function. We then select a subset of token representations from $\widehat{\boldsymbol{Z}}$ with $\sigma_\lambda(\langle \boldsymbol{u}_0, \boldsymbol{z}_{j,i}^\ell \rangle) > 0$. which correspond to non-zero projection values after thresholding, and we denote this subset as $\widehat{\boldsymbol{Z}}_s \subseteq \widehat{\boldsymbol{Z}}$. This selection step is used to remove the background [35]. We then compute the first three PCA components of $\widehat{\boldsymbol{Z}}_s$ with the first three eigenvectors of matrix $\widehat{\boldsymbol{Z}}_s^*\widehat{\boldsymbol{Z}}_s$ denoted as $\{\boldsymbol{u}_1, \boldsymbol{u}_2, \boldsymbol{u}_3\}$. We define the RGB tuple for each token as:

$$[r_{j,i}, g_{j,i}, b_{j,i}] = [\langle \boldsymbol{u}_1, \boldsymbol{z}_{j,i}^\ell \rangle, \langle \boldsymbol{u}_2, \boldsymbol{z}_{j,i}^\ell \rangle, \langle \boldsymbol{u}_3, \boldsymbol{z}_{j,i}^\ell \rangle], \quad i \in [N], j \in [J], \boldsymbol{z}_{j,i}^\ell \in \widehat{\boldsymbol{Z}}_s. \tag{12}$$

Next, for each image $\boldsymbol{X}_j$ we compute $\boldsymbol{R}_j, \boldsymbol{G}_j, \boldsymbol{B}_j$, where $\boldsymbol{R}_j = [r_{j,1}, \ldots, r_{j,N}]^\top \in \mathbb{R}^{d \times 1}$ (similar for $\boldsymbol{G}_j$ and $\boldsymbol{B}_j$). Then we reshape the three matrices into $\sqrt{N} \times \sqrt{N}$ and visualize the "PCA components" of image $\boldsymbol{X}_j$ via the RGB image $(\boldsymbol{R}_j, \boldsymbol{G}_j, \boldsymbol{B}_j) \in \mathbb{R}^{3 \times \sqrt{N} \times \sqrt{N}}$.

The PCA visualization of ViTs are evaluated similarly, with the exception of utilizing the "Key" features $\widehat{\boldsymbol{z}}_{j,i}^{\ell} = [(\boldsymbol{K}_1^* \widehat{\boldsymbol{z}}_{j,i}^{\ell})^\top, \ldots, (\boldsymbol{K}_K^* \widehat{\boldsymbol{z}}_{j,i}^{\ell})^\top]^\top$. Previous work [2] demonstrated that the "Key" features lead to less noisy space structures than the "Query" features. In the experiments (such as in Figure 3), we set the threshold value $\lambda = \frac{1}{2}$.

## B.3. Segmentation Maps and mIoU Scores

We now discuss the methods used to compute the segmentation maps and the corresponding mean-Union-over-Intersection (mIoU) scores.

Indeed, suppose we have already computed the attention maps $\boldsymbol{A}_k^\ell \in \mathbb{R}^N$ for a given image as in Appendix B.1. We then *threshold* each attention map by setting its top $P = 60\%$ of entries to $1$ and setting the rest to $0$. The remaining matrix, say $\tilde{\boldsymbol{A}}_k^\ell \in \{0,1\}^N$, forms a segmentation map corresponding to the $k^{\text{th}}$ head in the $\ell^{\text{th}}$ layer for the image.

Suppose that the tokens can be partitioned into $M$ semantic classes, and the $m^{\text{th}}$ semantic class has a boolean ground truth segmentation map $\boldsymbol{S}_m \in \{0,1\}^N$. We want to compute the quality of the attention-created segmentation map above, with respect to the ground truth maps. For this, we use the mean-intersection-over-union (mIoU) metric [28] as described in the sequel. Experimental results yield that the heads at a given layer correspond to different semantic features. Thus, for each semantic class $m$ and layer $\ell$, we attempt to find the best-matched head at layer $\ell$ and use this to compute the intersection-over-union, obtaining

$$\text{mIoU}_m^\ell \doteq \max_{k \in [K]} \frac{\|\boldsymbol{S}_m \odot \boldsymbol{A}_k^\ell\|_0}{\|\boldsymbol{S}_m\|_0 + \|\boldsymbol{A}_k^\ell\|_0 - \|\boldsymbol{S}_m \odot \boldsymbol{A}_k^\ell\|_0}, \tag{13}$$

where $\odot$ denotes element-wise multiplication and $\|\cdot\|_0$ counts the number of nonzero elements in the input vector (and since the inputs are boolean vectors, this is equivalent to counting the number of 1's). To report the overall mIoU score for layer $\ell$ (or without referent, for the last layer representations), we compute the quantity

$$\text{mIoU}^\ell \doteq \frac{1}{M} \sum_{m=1}^{M} \text{mIoU}_m^\ell, \tag{14}$$

and average it amongst all images for which we know the ground truth.

## B.4. MaskCut

We apply the MaskCut pipeline (Algorithm 4) to generate segmentation masks and detection bounding box (discussed in Section 3.2). As described by Wang et al. [46], we iteratively apply Normalized Cuts [42] on the patch-wise affinity matrix $\boldsymbol{M}^\ell$, where $\boldsymbol{M}_{ij}^\ell = \sum_{k=1}^{K} \langle \boldsymbol{U}_k^{\ell*} \boldsymbol{z}_i^\ell, \boldsymbol{U}_k^{\ell*} \boldsymbol{z}_j^\ell \rangle$. At each iterative step, we mask out the identified patch-wise entries on $\boldsymbol{M}^\ell$. To obtain more fine-grained segmentation masks, MaskCut employs Conditional Random Fields (CRF) [24] to post-process the masks, which smooths the edges and filters out unreasonable masks. Correspondingly, the detection bounding box is defined by the rectangular region that tightly encloses a segmentation mask.

---

**Algorithm 4** MaskCut

---

**Hyperparameter:** $n$, the number of objects to segment.
1: **function** MASKCUT($\boldsymbol{M}$)
2:     **for** $i \in \{1, \ldots, n\}$ **do**
3:         mask $\leftarrow$ NCUT($\boldsymbol{M}$)                                    ▷ mask is a boolean array
4:         $\boldsymbol{M} \leftarrow \boldsymbol{M} \odot$ mask                  ▷ Equivalent to applying the mask to $\boldsymbol{M}$
5:         masks[$i$] $\leftarrow$ mask
6:     **end for**
7:     **return** masks
8: **end function**

---

Following the official implementation by Wang et al. [46], we select the parameters as $n = 3, \tau = 0.15$, where $n$ denotes the expected number of objects and $\tau$ denotes the thresholding value for the affinity matrix $\boldsymbol{M}^\ell$, i.e. entries smaller than $0.15$ will be set to $0$. In Table 1, we remove the post-processing CRF step in MaskCut when comparing different model variants.

# C. Experimental Setup and Additional Results

In this section, we provide the experimental setups for experiments presented in Section 3 and Section 4, as well as additional experimental results. Specifically, we provide the detailed experimental setup for training and evaluation on Appendix C.1. We then present additional experimental results on the transfer learning performance of CRATE when pre-trained on ImageNet-21k [10] in Appendix C.2. In Appendix C.3, we provide additional visualizations on the emergence of segmentation masks in CRATE.

## C.1. Setups

**Model setup** We utilize the CRATE model as described by Yu et al. [51] at scales -S/8 and -B/8. In a similar manner, we adopt the ViT model from Dosovitskiy et al. [11] using the same scales (-S/8 and -B/8), ensuring consistent configurations between them. One can see the details of CRATE transformer in Appendix A.

**Training setup** All visual models are trained for classification tasks (see Section 2.2) on the complete ImageNet dataset [10], commonly referred to as ImageNet-21k. This dataset comprises 14,197,122 images distributed across 21,841 classes. For training, each RGB image was resized to dimensions $3 \times 224 \times 224$, normalized using means of $(0.485, 0.456, 0.406)$ and standard deviations of $(0.229, 0.224, 0.225)$, and then subjected to center cropping and random flipping. We set the mini-batch size as 4,096 and apply the Lion optimizer [8] with learning rate $9.6 \times 10^{-5}$ and weight decay 0.05. All the models, including CRATEs and ViTs are pre-trained with 90 epochs on ImageNet-21K.

**Evaluation setup** We evaluate the coarse segmentation, as detailed in Section Section 3.2, using attention maps on the PASCAL VOC 2012 *validation set* [13] comprising 1,449 RGB images. Additionally, we implement the MaskCut [46] pipeline, as described in Section 3.2, on the COCO *val2017* [27], which consists of 5,000 RGB images, and assess our models' performance for both object detection and instance segmentation tasks. *All evaluation procedures are unsupervised, and we do not update the model weights during this process.*

## C.2. Transfer Learning Evaluation

We evaluate transfer learning performance of CRATE by fine-tuning models that are pre-trained on ImageNet-21k for the following downstream vision classification tasks: ImageNet-1k [10], CIFAR10/CIFAR100 [25], Oxford Flowers-102 [33], Oxford-IIIT-Pets [37]. We also finetune on two pre-trained ViT models (-T/8 and -B/8) for reference. Specifically, we use the AdamW optimizer [29] and configure the learning rate to $5 \times 10^{-5}$, weight decay as 0.01. Due to memory constraints, we set the batch size to be 128 for all experiments conducted for the base models and set it to be 256 for the other smaller models. We report our results in Table 3.

| Datasets | CRATE-T | CRATE-S | CRATE-B | ViT-T | ViT-B |
|---|---|---|---|---|---|
| # parameters | 5.74M | 14.12M | 38.83M | 10.36M | 102.61M |
| ImageNet-1K | 62.7 | 74.2 | 79.5 | 71.8 | 85.8 |
| CIFAR10 | 94.1 | 97.2 | 98.1 | 97.2 | 98.9 |
| CIFAR100 | 76.7 | 84.1 | 87.9 | 84.4 | 90.1 |
| Oxford Flowers-102 | 82.2 | 92.2 | 96.7 | 92.1 | 99.5 |
| Oxford-IIIT-Pets | 77.0 | 86.4 | 90.7 | 86.2 | 91.8 |

**Table 3:** Top 1 accuracy of CRATE on various datasets with different model scales when pre-trained on ImageNet-21K and fine-tuned on the given dataset.

## C.3. Additional Visualizations

*Supervised CRATE*

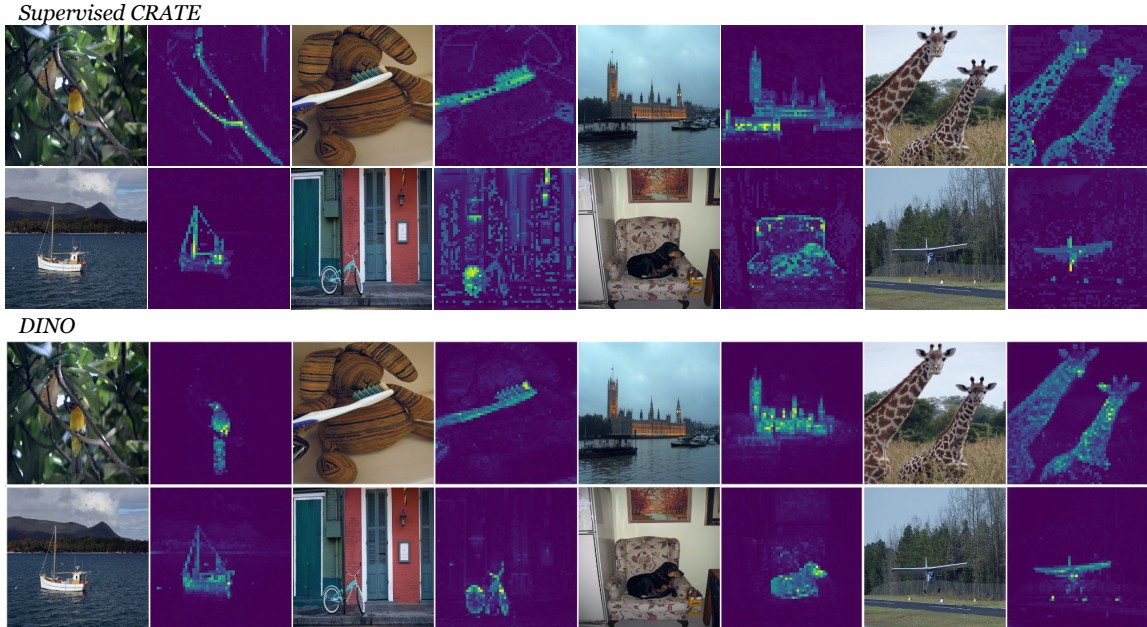

**Figure 8: Additional visualizations of the attention map of CRATE-S/8 and comparison with DINO [6].** *Top 2 rows*: visualizations of attention maps from *supervised* **CRATE**-S/8. *Bottom 2 rows*: visualizations of attention maps borrowed from DINO's paper. The figure shows that *supervised* **CRATE** has at least comparable attention maps with DINO. Precise methodology is discussed in Appendix B.1.

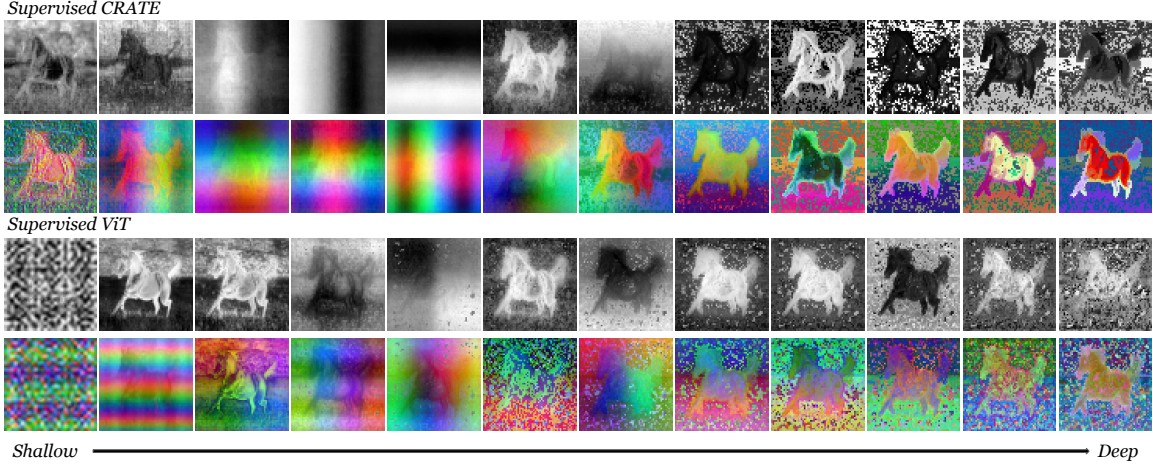

**Figure 9: Additional layer-wise PCA visualization.** *Top 2 rows*: visualizations of the PCA of the features from *supervised* **CRATE**-B/8. *Bottom 2 rows*: visualizations of the PCA of the features from *supervised* ViT-B/8. The figure shows that *supervised* **CRATE** shows a better feature space structure with an explicitly-segmented foreground object and less noisy background information. The input image is shown in Figure 1's *top left* corner. Precise methodology is discussed in Appendix B.2.

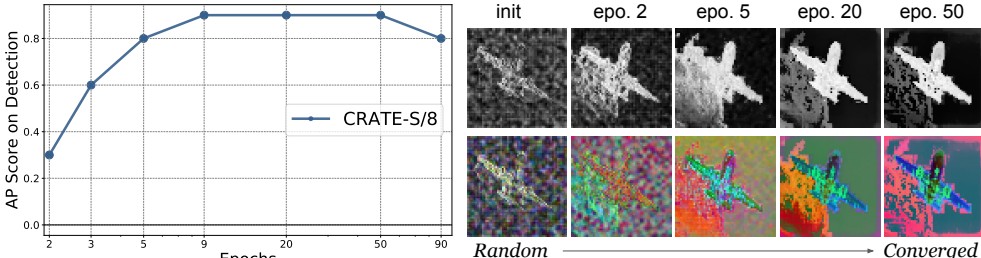

**Figure 10: Effect of training epochs in supervised CRATE.** (*Left*) Detection performance *computed at each epoch* via MaskCut pipeline on COCO val2017 (Higher AP score means better detection performance). (*Right*) We visualize the PCA of the features at the penultimate layer *computed at each epoch*. As training epochs increase, foreground objects can be explicitly segmented and separated into different parts with semantic meanings.

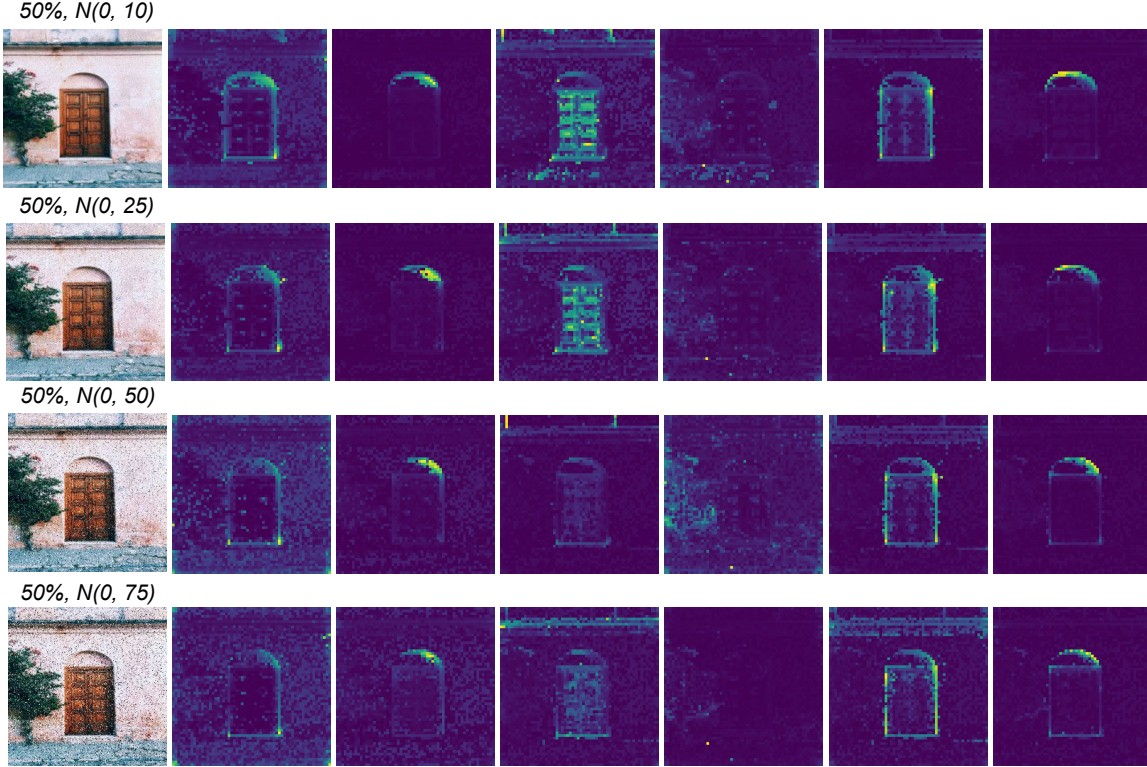

**Figure 11: Adding Gaussian noise with different standard deviation.** We add Gaussian noise to the input image on a randomly chosen set of 50% of the pixels, with different standard deviations, and visualize all 6 heads in layer 10 of **CRATE**-S/8. The values of each entry in each color of the image are in the range $(0, 255)$. The *right 2 columns*, which contain edge information, remain unchanged with different scales of Gaussian noise. The *middle column* shows that texture-level information will be lost as the input becomes noisier.

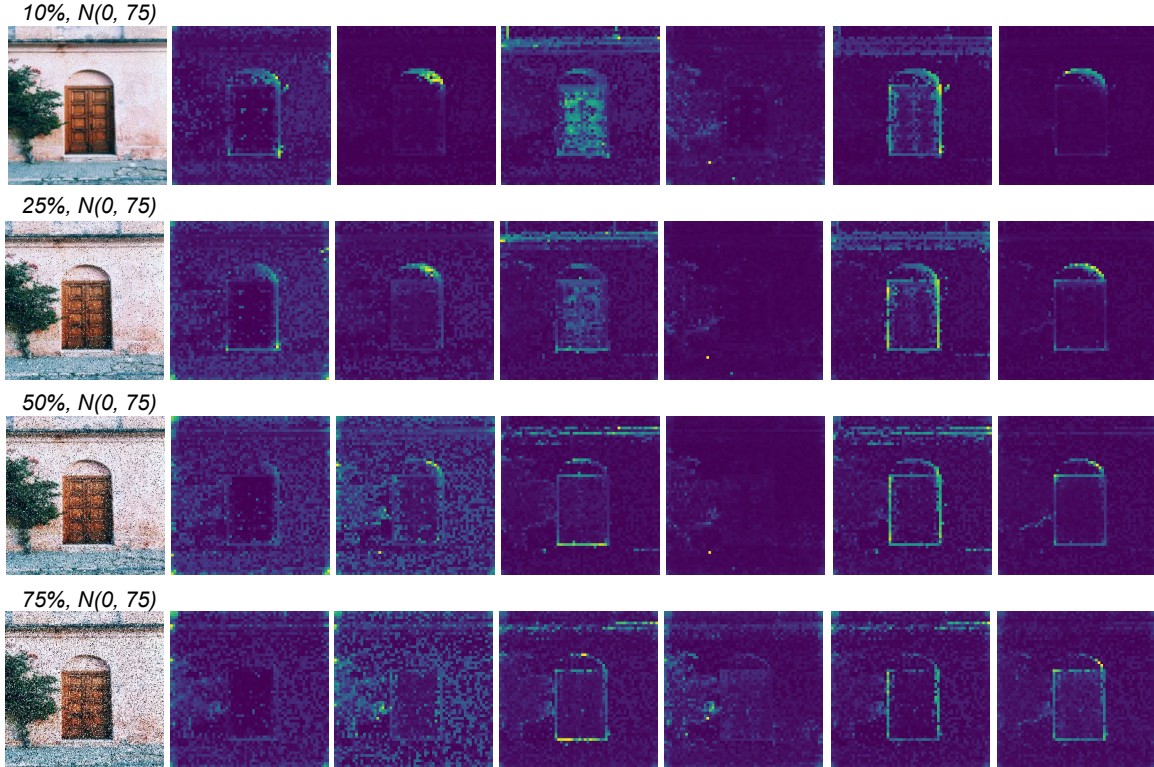

**Figure 12: Adding Gaussian noise to a different percentage of the pixels.** We add Gaussian noise with standard deviation 75 to a randomly chosen set of pixels within the input image, taking a different number of pixels in each experiment. We visualize all 6 heads in layer 10 of **CRATE**-S/8. The values of each entry in each channel of the image are in the range $(0, 255)$. In addition to the observation in Figure 11, we find that **CRATE** shifts its focus as the percentage of noisy pixels increases. For example, in the middle column, the head first focuses on the texture of the door. Then, it starts to refocus on the edges. Interestingly, the tree pops up in noisier cases' attention maps.

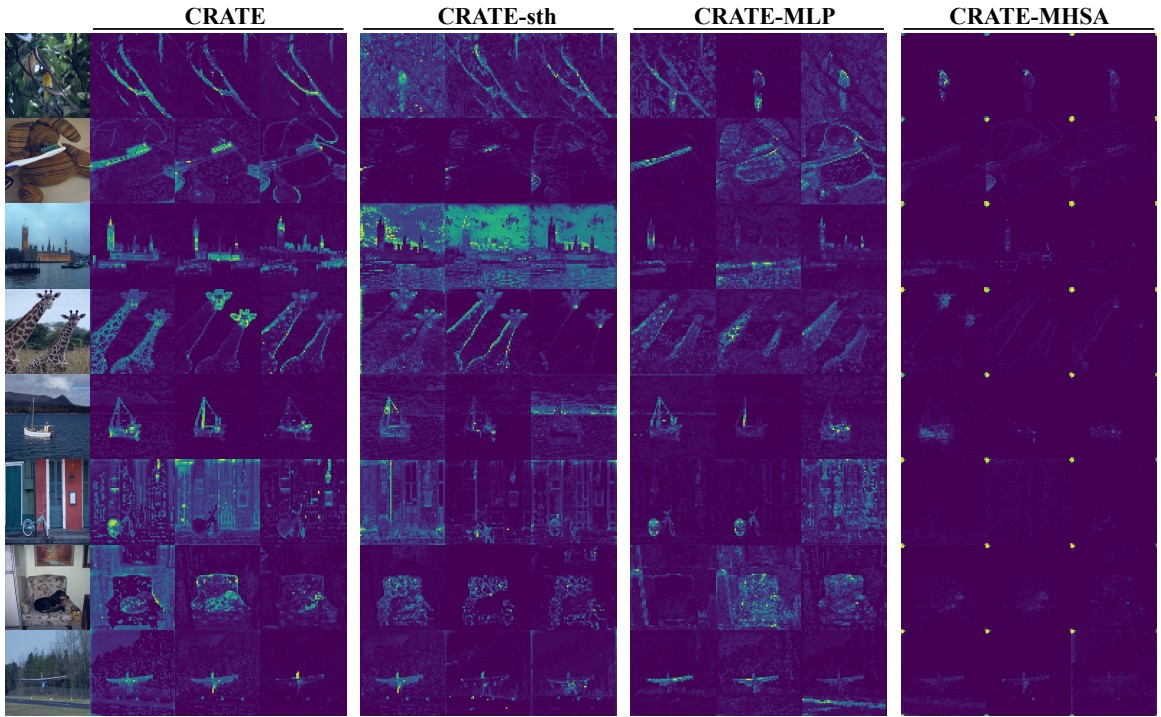

**Figure 13: Attention map of CRATE's variants in second-to-last layer**. In addition to the quantitative results discussed in Section 4, we provide visualization results for the architectural ablation study. **CRATE-MLP** and **CRATE-MHSA** have been discussed in Section 4 while **CRATE-sth** maintains both `MSSA` and `ISTA` blocks, and instead switches the activation function in the `ISTA` block from ReLU to soft thresholding, in accordance with an alternative formulation of the `ISTA` block which does not impose a non-negativity constraint in the LASSO (see Section 2.1 for more details). Attention maps with clear segmentations emerge in all architectures with the `MSSA` block.

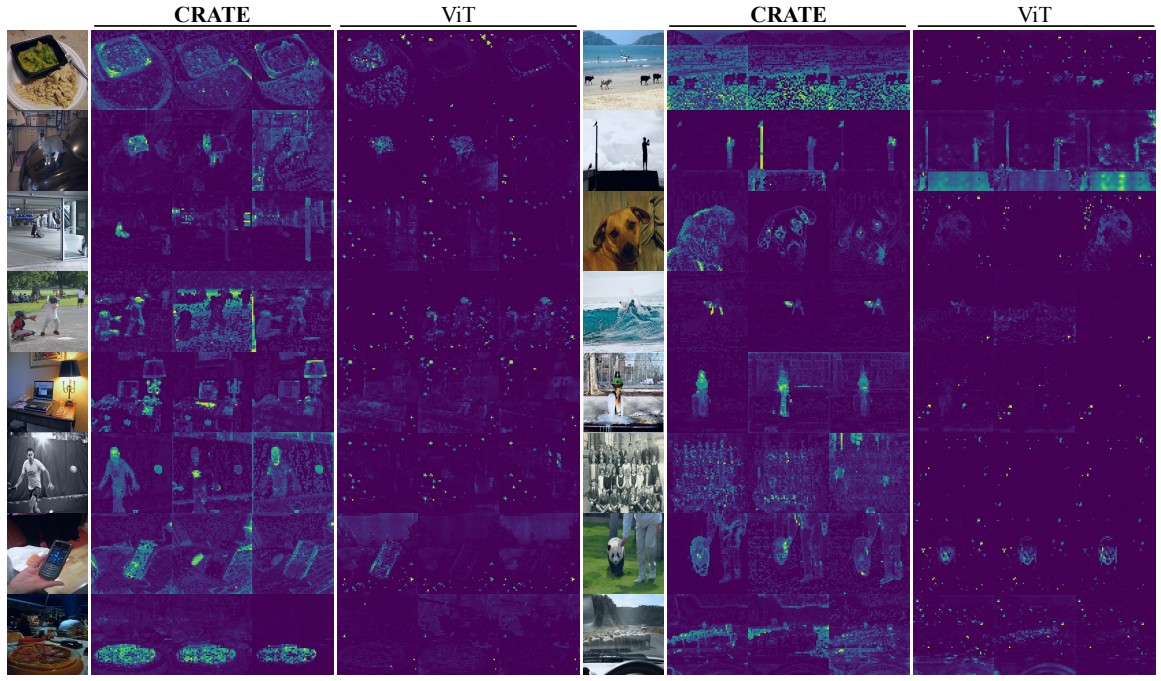

**Figure 14: More attention maps of supervised CRATE and ViT** on images from COCO *val2017*. We select the second-to-last layer attention maps for **CRATE** and the last layer for ViT.

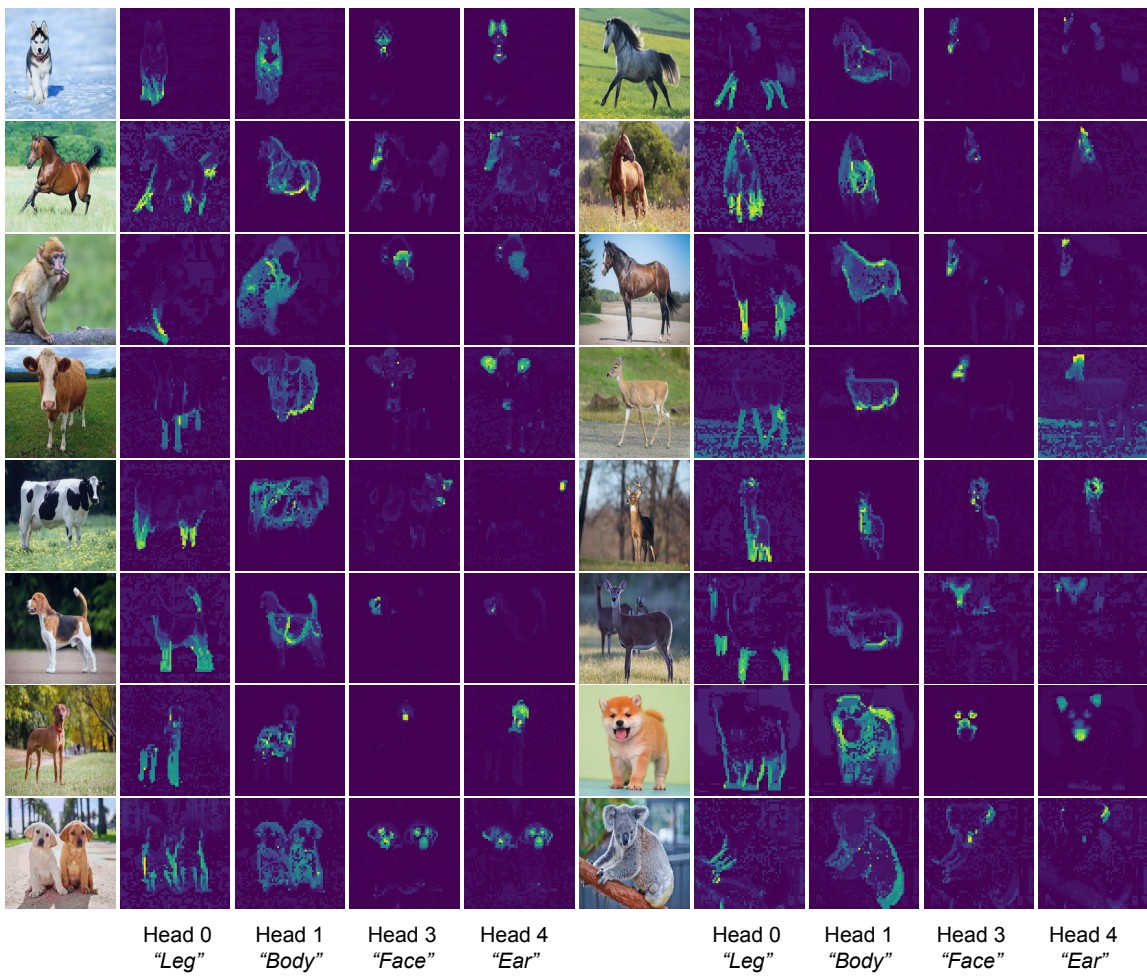

| Head 0
*"Leg"* | Head 1
*"Body"* | Head 3
*"Face"* | Head 4
*"Ear"* | | Head 0
*"Leg"* | Head 1
*"Body"* | Head 3
*"Face"* | Head 4
*"Ear"* |

**Figure 15: More Visualization of semantic heads.** We forward a mini-batch of images through a supervised CRATE and examine the attention maps from all the heads in the penultimate layer. We visualize a selection of attention heads to show that certain heads convey specific semantic meaning, i.e. *head 0 ↔ "Legs", head 1 ↔ "Body", head 3 ↔ "Face", head 4 ↔ "Ear"*.

