# OpenReview forum: "Emergence of Segmentation with Minimalistic White-Box Transformers"
_CPAL.cc/2024/Conference — CPAL 2024 (Proceedings Track) Oral_

### Official Review · Reviewer_xxKA · 2023-09-28
**An Official Review about Acceptance**

**Rating:** 7
**Confidence:** 4

**Review:**

# Summary

By conducting thorough quantitative and qualitative analyses, the paper effectively highlights the prowess of CRATE models in semantic segmentation, achieved through simple supervised training of classification. The research strongly underscores the merits of White-Box Transformers.

# Strengths

1. **Well-Written**: The paper is structured lucidly, with detailed experimental settings, making it accessible and easy to follow.
2. **Robust Experiments**: This paper undertakes comprehensive quantitative and qualitative evaluations, attesting to CRATE's superiority.
3. **Interesting Findings**: The study underscores the importance of a more interpretable architecture design for the academic community.

# Questions

1. In Figure 6 (left), VIT-B displays optimal performance in the last block, whereas CRATE-B peaks in the penultimate layer. Paper [1] suggests that the penultimate-layer features in ViTs trained with DINO strongly correlate with visual input saliency. Considering both models in Figure 6 are supervised-trained, why do they peak in different layers? Is there a more cogent explanation?
2. CRATE advances semantic segmentation via its white-box design. How does it stack up against other black-box architectural designs, such as PVT [2]? Can black-box designs also enhance the emergence?

# References

[1] Emerging properties in self-supervised vision transformers

[2] Pyramid vision transformer: A versatile backbone for dense prediction without convolutions

---

### Official Review · Reviewer_WH3x · 2023-10-06
**A interesting work of a novel and explainable architecture of Vision Transformer**

**Rating:** 6
**Confidence:** 4

**Review:**

This paper proposes a white-box Transformer architecture for visual segmentation tasks. The model is designed to optimize the sparse rate reduction objective, which results in the property that each layer first compresses the distribution of tokens and then sparsely encodes the next representation. The visualization shows a good segmentation performance compared with ViT. Meanwhile, each layer and attention head is explainable. This paper is very well-written. However, I still have several questions or concerns.

1. What is the major contribution compared with [51]? From my understanding, the architecture seems similar to [51]. Is the contribution mainly about the tasks on segmentation?

2. It would be better if there were more theoretical explanations of the relationship between the CRATE model architecture and segmentation tasks in Section 2. I like the proposed mechanism of each layer, but it is unclear how it helps segmentation tasks in theory.

3. I will treat such work as an important work on the theoretical understanding of Vison Transformers. The proposed mechanism of first compressing and then sparsely encoding is very interesting to me. Some recent theoretical works [a], [b], [c] on (Vision) Transformers provide another explanation of the learning process, which could be summarized as feature matching and selection. It would be great if this paper could cover a discussion with these works.

4. A minor point. How do you compare the training efficiency of your proposed method with existing works on segmentation tasks?


[a] S. Jelassi et al., Neurips 2022. "Vision transformers provably learn spatial structure."

[b] H. Li et al., ICLR 2023. "A Theoretical Understanding of Shallow Vision Transformers: Learning, Generalization, and Sample Complexity."

[c] Y. Li et al., ICML 2023. "How do transformers learn topic structure: Towards a mechanistic understanding."

---

### Official Review · Reviewer_KcjJ · 2023-10-07
**Captivating observation about the emergence of segmentation properties**

**Rating:** 7
**Confidence:** 4

**Review:**

This paper investigates the emergence of segmentation properties within Vision Transformer models (ViTs). Contrary to the prevailing belief that segmentation properties predominantly result from intricate self-supervised techniques like DINO, this paper illustrates that these properties can also manifest through architectural choices within the conventional supervised training paradigm.

The authors provide a comprehensive overview of the background literature necessary for grasping the paper's content, rendering their work self-contained and accessible. Moreover, the paper carefully examines both qualitative and quantitative metrics to support its claims. Specifically, the authors demonstrate that their findings hold across diverse datasets, varying model sizes, and a spectrum of evaluation metrics. The inclusion of insightful ablation studies, notably the architectural modification of ViT (specifically, the replacement of MHSA with MSSA), adds depth to the analysis. A notable feature of this paper lies in its detailed description of the experimental methodologies, thoughtfully provided in the appendix (huge plus!). This greatly improves the ability to replicate the results and encourages their use as a basis for subsequent research endeavors.

In conclusion, the findings presented in this work hold substantial value for the research community. They introduce and validate novel and intriguing insights previously unexplored, effectively challenging established beliefs regarding the emergence of segmentation properties in ViTs.

A few minor comments and questions for the authors:

1. Could you elaborate more on the thought process that led you to investigate this specific architecture choice? Usually, papers introducing novel architectures lack insight into this aspect, and such findings may appear as if they came out of nowhere. However, I believe that architectural improvements are typically the result of an iterative process, often involving failed attempts. Including a paragraph describing other architectural options you explored (if any) and explaining how and where they fell short in producing segmentation properties would be highly beneficial. This information could prove invaluable to fellow researchers seeking to build upon your work, potentially saving them time and effort by avoiding similar pitfalls.

2. Your paper demonstrates the emergence of segmentation in the attention maps of CRATE trained in a supervised manner on ImageNet-21k. Have you observed these same segmentation properties persisting after fine-tuning the model for other downstream tasks, or do these features tend to diminish during the transfer-learning process? It would be valuable to include an analysis of some of the transfer-learning datasets mentioned in Appendix C.2 to shed light on this aspect.

3. Figure 11. has a typo in the x-axis label: “Epocs” -> “Epochs”

4. Looking at Figure 11 (left), it’s evident that AP score saturates after 9th epoch. Could you provide the same analysis as in  Figure 11 (left and right) for at least one more model and one more dataset? This additional data would provide insight into when these segmentation properties typically manifest during training. Are they consistently present early in training, or does their appearance depend on factors like model size and dataset selection?

5. Could you also provide the same analysis as in Figure 11 for classic ViT models. This would offer further insights into whether classic ViTs could have benefited from extended training or if they also saturate early in training, albeit with notably lower scores.

---

### Meta-Review · Area_Chair_JySd · 2023-11-12

**Recommendation:** Accept (Poster)
**Confidence:** 4

**Metareview:**

This paper provides insightful investigations into whether an effective segmentation transformer-based model can come from supervised training/pre-training. Detailed model and optimization designs are offered, together with extensive analysis and informative visualizations. The authors also conduct great efforts during the rebuttal period. All reviewers voted for the acceptance of this submission and I agree it can benefit our community.

---

### Decision · Program_Chairs · 2023-11-19

**Decision:**

Accept (Oral)

**Comment:**

This paper demonstrated the emerging capability of segmentation in a white-box transformer-like architecture known as CRATE, with minimalistic supervised training. The paper has been well received by the reviewers, and may inspire future works on the design of white-box foundation models that are strongly performant.

The action PC chair for this paper is Yuejie Chi, who made the decision after carefully reading the paper as well as the comments by all reviewers and AC. The decision is agreed by all PC chairs.